# Estimation of the Influence of a Noisy Environment on the Binary Decision Strategy in a Quantum Illumination Radar

**DOI:** 10.3390/s22134821

**Published:** 2022-06-25

**Authors:** Sylvain Borderieux, Arnaud Coatanhay, Ali Khenchaf

**Affiliations:** Lab-STICC, UMR, CNRS 6285, ENSTA Bretagne, 2 Rue François Verny, 29 806 Brest, France; arnaud.coatanhay@ensta-bretagne.fr (A.C.); ali.khenchaf@ensta-bretagne.fr (A.K.)

**Keywords:** quantum illumination, entanglement, quantum discord, binary decision strategy, quantum information theory

## Abstract

A quantum illumination radar uses quantum entanglement to enhance photodetection sensitivity. The entanglement is quickly destroyed by the decoherence in an environment, although the sensitivity enhancement could survive thanks to quantum correlations beyond the entanglement. These quantum correlations are quantified by the quantum discord. Here, we use a toy model with an amplitude damping channel and Lloyd’s binary decision strategy to highlight the possible role of these correlations from the perspective of a quantum radar.

## 1. Introduction

Since the late 20th Century, radar technology has been used in many applications, especially for maritime and aeronautic purposes [1,2,3]. One of the most important subjects in radar technology concerns the detection of stealth targets in the context of background noise. In another way, the current development of quantum technologies provides new possibilities for remote detection, leading to the concept of quantum radar. This paper presents a “toy model” for quantum radar based on quantum entanglement between pairs of photons. Such a simple model does not aim to be realistic but rather provides pedagogical value concerning the potentiality created by the quantum radar.

The current development of quantum technologies for the transmission of information introduced the idea of “quantum radar”, although this idea remained of little interest until Lloyd’s article was published in 2008 [4]. In this article, Seth Lloyd showed that the quantum entanglement with pairs of photons can significantly improve the remote detection sensitivity in the optical frequency regime. This way of using entanglement for remote detection is called “quantum illumination” (QI). Since this article, interest in the field of quantum radar has grown. New theoretical and experimental research has been conducted on this subject [5,6,7,8,9,10,11,12]. Research around the quantum radar has moved from focusing on the individual photons to small bunches of photons [4,11]. In the same sense, research has moved from the optical frequency regime [4] to the microwave frequency regime [11,12,13], which is more suitable for radar applications but also more challenging. In this context, new technologies are currently being developed to make quantum illumination possible in the microwave regime. For instance, we can cite the Josephson junction, which enables the direct production of microwave-entangled photons at low temperature. There is also the coupling between an optical photon and a microwave photon [11]. Then, the Nitrogen-Vacancy centers (called NV centers) also permit the production of microwave entangled photons. Despite the great difficulties relating to the feasibility of such a quantum radar, this research field is highly active.

The quantum radar has the same purpose as the conventional radar, but the functioning relies on the principles of quantum mechanics.

We recall that a conventional radar works with the classical theory of electromagnetism based on Maxwell equations. To briefly summarize it, a radar is a device that sends an electromagnetic wave to detect a reflecting object that reflects a fraction of the incident wave to the radar. The radar scheme is characterized by the energy ratio called the “radar equation” between the received and the emitted wave. The received wave provides information on the detection and the position of an object (i.e., radar target) using signal processing.

In comparison, a quantum radar relies on quantum mechanics to work, but the current definition is not as clear. A quantum radar could be defined as a conventional radar that uses only a quantum electronic device to improve its sensitivity. Hence, if any quantum electronic device is found in the radar system, we could refer to it as a quantum radar. From another perspective, the use of a low number of photons instead of classical electromagnetic waves also means that we can refer to a quantum radar. Using only one of these aspects theoretically means that we can refer to a quantum radar. Given these arguments, the current definition of a quantum radar is still somewhat ambiguous and unclear, but the common thread between these definitions is the requirement of quantum mechanics [11,14].

Among the known types of quantum radar, the quantum illumination radar proposed by Lloyd [4] remains the most interesting because the enhancement of the sensitivity of the radar from the use of quantum entanglement is significant. Below, the quantum radar exclusively refers to the QI radar. Lloyd’s quantum illumination is based on the use of pairs of entangled photons, whereby one photon is trapped in the radar system, while the second photon is emitted into space to be reflected by a target. The emitted photon is reflected by the target, and it comes back to the radar to be measured. In the radar, the detection strategy relies on a joint measurement of the pair of photons. This is a striking point that is nevertheless challenging in experimental terms, as specified in the current literature [11,12].

In Lloyd’s article [4], the sensitivity enhancement induced by the entanglement could be maintained even if the entanglement is quickly lost during the propagation phase in the optical regime. That means some quantum correlations, i.e., some quantum information, could survive to the decoherence induced by the propagation environment which destroys the initial entanglement. The entanglement phenomenon underlines the presence of strong quantum correlations, but they are quickly lost due to the environment. The sensitivity resilience could be explained by quantum correlations beyond the entanglement [10]. Such correlations can be quantified by the quantum discord [15]. There is currently lack of studies examining the environmental influence (such as the atmosphere) on the quantum correlation evolution in the quantum illumination radar.

In this paper, we work with a toy model with Lloyd’s quantum illumination scheme, introducing the damping effect of the propagation environment to follow the evolution of quantum information. The objective consists in studying a simple QI radar from the point of view of the quantum information theory introducing the quantum discord. Therefore, we study the quantum illumination radar for a pair of qubits entangled on thermal energy states since the detection process is photodetection. A generalized amplitude damping quantum channel is used to model the environmental influence acting on the thermal energy levels of one qubit alone. We use Lloyd’s decision strategy to introduce a low average number $n_{b}$ of thermal photons by mode, representing the environmental thermal noise. The decision strategy is linked to the quantum channel through a damping probability *p* from the channel model [16]. For the sake of simplicity, we have limited ourselves to a low number of thermal modes d=2 corresponding to qubit states widely used in quantum information theory [17,18]. The case d=2 is a limiting case for small energy states, so our toy model does not aim to represent a realistic case.

This paper is divided into four sections. Section 2 explains the basis of the quantum illumination radar and the need to adopt a quantum formalism. The decision strategy is adapted to account for the environmental influence using an amplitude-damping quantum channel acting like a heat bath. In Section 3, we study the radar under the scope of quantum information theory to follow the evolution of information. Next, in Section 4, we introduce Lloyd’s binary decision strategy according to the quantum channel used. We discuss the information evolution in the quantum channel linked to the decision theory. Section 5 provides the conclusion.

## 2. Description of a Quantum Illumination Radar

In this section, we introduce Lloyd’s QI radar for pairs of entangled photons. We use a conventional radar scheme to explain the differences between the QI radar and a classical radar. The quantum formalism for the QI radar is introduced in Section 2.2. Then, the formalism used to define the amplitude damping channel is given in Section 2.3.

### 2.1. Description of the Radar System

The principle of a QI radar relies on the entanglement and on a joint measurement on the entangled system [4]. We start by considering a conventional radar scheme to highlight the differences between a quantum radar and a classical radar.

The QI radar scheme is built like the conventional radar scheme in a monostatic case. We have four main sections that permit us to define a quantum radar equation. In conventional radar theory, the radar scheme is constituted by the emitter (an emitting antenna), the propagation channel (the atmosphere), the target (a reflecting object), and the receiver (a receiving antenna) [1]. Following the same scheme, the quantum radar scheme is constituted by the emitter, which produces pairs of entangled photons in optical or microwave frequencies [11]. Next, the propagation channel describes the perturbative propagation medium, such as the atmosphere, acting on the emitted photon. The second photon is trapped in the radar, which is a non-perturbative medium. The radar target is an object that reflects the incident photon without absorbing it. Then, the receiver is a quantum sensor taking a joint measurement of the photon received and the trapped photon.

Both radar schemes are used to define the classical radar equation in Equation (Equation 1) and the quantum radar equation in Equation (Equation 2).

In classical radar theory, electromagnetic waves (EW) are used to detect distant objects. The energy used to produce the EW has to be significant, as the atmosphere induces an attenuation of the incident wave. This attenuation is due to scattering and absorption by the atmospheric molecules (H${}_{2}$, O${}_{2}$, etc.) [1]. Furthermore, an energy decay occurs because the EW propagates in all directions. Only a fraction of the incident wave is reflected by the target in all directions before it is received by the radar antenna. Therefore, the classical radar equation written in Equation (Equation 1) is an energy ratio between the received wave and the emitted wave.
(1)$$E_{r}=\frac{P_{t}\tau G_{t}}{4\pi R^{2}}\times \frac{1}{L^{2}}\times \frac{\sigma_{C}}{4\pi R^{2}}\times A_{e}\;\left[\mathrm{energy}\right]$$

In Equation (Equation 1), $P_{t}$ is the transmitted power over a time τ with the gain in the antenna $G_{t}$. The factor $4\pi R^{2}$ shows that the incident wave is irradiated in all directions of the space in a sphere of radius *R*. The factor $1/L^{2}$ represents the wave attenuation due to the absorption and scattering over a round -trip. $\sigma_{C}$ is the Radar Cross Section (RCS) that indicates how an object reflects an incident wave. The RCS depends on the target geometry and on its nature [1]. $A_{e}=G_{\mathrm{rc}}\lambda^{2}/\left(4\pi \right)$ is the effective area of the receiving antenna with a gain $G_{\mathrm{rc}}$, and it depends on the wavelength λ=c/f [3]. Here, the classical radar equation has the form of a signal-to-noise ratio (SNR).

In quantum illumination radar theory, we emit single photons, not EW. In the atmosphere, the emitted photon can be absorbed or scattered, in turn implying the loss of information, i.e., the loss of any useful signal. Hence, we add a probability factor $P_{p}$ for the photon not being absorbed or scattered by the propagation environment; the emitted energy and the received energy are the same. In addition, this photon energy E=ℏω is far lower than the energy of classical EW. Therefore, the quantum radar equation is not an energy ratio but a probability law to retrieve the emitted photon after its propagation and its reflection on the target. This probability law is written in Equation (Equation 2) following the form of Equation (Equation 1).
(2)$$P_{r}=P_{p}^{2}\times \frac{\sigma_{Q}}{4\pi R^{2}}\times P_{a}\;\left[\mathrm{probability}\right]$$
where $P_{p}$ is the probability of the photon not being absorbed and scattered by atmospheric molecules along the propagation. This probability is squared since the photon makes a round-trip between the radar and the target. $\sigma_{Q}$ is the Quantum Radar Cross Section (QRCS), which is the analog equivalent of the RCS, i.e., the way for an object to reflect the incident photon. Some research has already been conducted on the QRCS between a photon and reflecting targets with canonical geometries [19]. For photons with polarization and a non-polarized target, the interaction does not significantly alter the quantum state of the photon [20]. $P_{a}$ is the probability of detection of the quantum sensor. It depends on the effective area of the sensor and on its sensitivity threshold. For instance, detecting microwave photons is more challenging than detecting optical photons because their energy is much lower.

It should be noted that the QI radar equation is equal to the single-photon radar equation described in Section 4 since, in both radars, only one photon is emitted. The fundamental difference lies in the entanglement for the QI radar between photon A (ancilla) trapped in the radar and photon S (signal) that propagates. In the following, we describe in Figure 1 how a QI radar works using entangled photonic qubits.

In Figure 1, the QI radar principle for a pair of photonic qubits is described in three steps. These steps condition the quantum formalism used in Section 2.2 and Section 2.3 and the decision theory used in Section 4.

Step 1 consists in creating the pair of entangled photonic qubits. Photon A (ancilla) is represented in blue and photon S (signal) is represented in red. Both are qubits, which means that we describe them with a state vector ${|\varphi \rangle }_{i=A,S}$ as a superposition of any two quantum states 0 and 1: ${|\varphi \rangle }_{i}=1/\sqrt{2}\left({|0\rangle }_{i}+{|1\rangle }_{i}\right)$. The state vector represents a quantum state φ of the quantum system i = A, S in a Hilbert space $H_{i}$ of dimension 2. The state vectors represent pure quantum states with a basis of vectors in $H_{i}$. However, in practice, we usually prefer the density matrix $\hat{\rho }=\sum_{j,k}p_{j,k}|j,k\rangle \langle j,k|$ on the eigenvectors $|j,k\rangle$ with the associated probability $p_{j,k}$. The density matrix notation is suitable to mixed states, and it is widely used in quantum information theory because a density operator ${\hat{\rho }}_{i}$ represents all the available information about a quantum state on the system “i” [17] (see calculations in Section 3.1).

In Step 2, the radar spatially separates the pair of photonic qubits. Photon A is kept in the radar. Photon S is emitted into space towards a reflecting object.

In Step 2a, photon A is trapped in the radar using a mirror cavity without any loss. This is an ideal case because ensuring a lossless cavity is very difficult in practice. Another point is to prevent any perturbation in system A, particularly to prevent any measurement.

In Step 2b, photon S travels through a propagation environment, and the quantum state of S is perturbed by the interaction with the environment. For instance, in the atmosphere, the perturbations are mainly due to atmospheric molecules (N${}_{2}$, O${}_{2}$, etc.). The perturbations undergone by photon S have an impact on the system AS since the system is initially entangled. After the propagation, photon S interacts with the reflecting object. We assume the photon is simply reflected by the object before it comes back to the radar. This approach is idealistic since in practice photon S can be absorbed or scattered during the propagation; furthermore, it could be also reflected anywhere by the object. Finally, photon S is caught by the receiver of the radar.

Both photons are gathered in the radar system in Step 3. The QI radar takes a joint measurement on system AS to identify its new quantum state. In reference to Lloyd’s quantum illumination [4], the initial entanglement is lost due to the propagation in the environment, and therefore the quantum state is modified.

Step 2b introduces two significant sources of perturbation independent from any technological devices used in the radar. In this step, there is a perturbation due to the propagation environment and a perturbation due to the interaction with a distant object.

The interaction between the photon and the object is calculated with the QRCS. According to the references [19,20], the interaction effect on the quantum state of the photon is negligible because the interaction time is too fast. Therefore, considering the object as a simple reflecting plate, we suppose this interaction effect is negligible.

The propagation environment has a strong effect on the photon state since it induces the decoherence, which in turn destroys the initial entanglement. In Lloyd’s article [4], the entanglement decay is immediate in the optical frequency regime.

In our toy model based on Lloyd’s quantum illumination, we have decided not to focus on the object influence on the qubit state. Instead, we have focused our attention on the propagation environment and its influence on the qubit state. This influence on the qubit state S affects the state of the system AS due to the quantum correlations. A generalized amplitude damping channel is used in Section 2.3 to model the environmental influence, and the link with the decision theory (Section 4) is made with the damping parameter *p* used in the quantum channel. The channel permits one to compute the quantum information evolution as a function of *p* in Section 3.2.

Lloyd’s quantum illumination is based on the photodetection of a target of reflectivity η with single-photon emissions in a bath of thermal photons. The thermal photons are assumed to have an average occupation probability $n_{b}=\left(1-e^{-\beta \hbar \omega }\right)e^{-\beta \hbar \omega }$ over the thermal modes, where $\beta ={\left(k_{B}T\right)}^{-1}$ with $k_{B}$, the Boltzmann constant, and *T* is the temperature. According to Lloyd’s article, the photodetector can distinguish $d=W\times t_{d}$ modes during a detection event where *W* is the frequency bandwidth and $t_{d}$ is the temporal window for the detection. According to the channel used in Section 2.3, we have limited ourselves to the limiting case of d=2 modes. The photodetector can detect at most one photon in a detection event that corresponds to $dn_{b}\ll 1$. This approximation is valid for optical frequency regimes, but this is not the case when we move on to microwave regimes. The binary decision strategy relies on photodetection, which is why we must work on thermal energy modes in the quantum damping channel to model the situation. The quantum channel depends on a damping parameter *p*, which is theoretically dependent on the environment but *p* does not represent the absorption/scattering probability $P_{p}$ of Equation (Equation 2). The decision strategy uses this parameter *p* to compose the probabilities of detection error and the signal-to-noise ratios in Section 4.

Thus, we must work with pairs of entangled thermal qubits in the quantum channel. The next section gives the formalism for the quantum states in the QI radar. Section 2.3 uses this formalism to define the amplitude-damping quantum channel.

### 2.2. Expression of the Quantum States in the QI Radar

Firstly, we discuss about the quantum entanglement and its consequences in the equations. Then, we introduce the quantum formalism for the photonic qubits and for the state of two entangled qubits.

The entanglement is a purely quantum phenomenon, and there is no classical counterpart [21]. A quantum system “1,2” in a quantum state ${\hat{\rho }}_{1,2}$ constituted by two subsystems “1” and “2” in quantum states ${\hat{\rho }}_{1}$ and ${\hat{\rho }}_{2}$ is entangled if we cannot describe it as a product state of its two subsystems ${\hat{\rho }}_{1,2}\neq {\hat{\rho }}_{1}⊗{\hat{\rho }}_{2}$, where ⊗ is the Kronecker product. A product state is also called a ”separable state”, i.e., a state with no entanglement. Therefore, an entangled state induces strong quantum correlations between subsystems “1” and “2”. This implies that the state ${\hat{\rho }}_{1}$ of the subsystem “1” is conditioned by the state ${\hat{\rho }}_{2}$ of the subsystem “2” and vice versa. The entanglement is the characteristic feature of quantum mechanics, and it represents the key resource for several information processes in quantum information theory [17,18]. Nevertheless, we have seen in Section 1 that some quantum correlations exist beyond entanglement. These correlations can be quantified by the quantum discord, which is different from the entanglement. The difference between the entanglement and the quantum discord is pointed out in Section 3.1.

Now, we must define the formalism used for thermal qubits in Equation (Equation 3) on the thermal energy states $|\varepsilon_{i=1,2}\rangle$, since Lloyd’s quantum illumination is based on the photodetection.
(3)$${\hat{\rho }}_{S,\mathrm{thermal}}=\frac{1}{2}\left(|\varepsilon_{1}\rangle \langle \varepsilon_{1}|+|\varepsilon_{2}\rangle \langle \varepsilon_{2}|\right)$$

Equation (Equation 3) is the reduced density operator on the subsystem S involved in the entangled state ${\hat{\rho }}_{\mathrm{AS},\mathrm{thermal}}$ in Equation (Equation 4) on the basis of the energy states $\left\{|\varepsilon_{1}\varepsilon_{1}\rangle ,|\varepsilon_{1}\varepsilon_{2}\rangle ,|\varepsilon_{2}\varepsilon_{1}\rangle ,|\varepsilon_{2}\varepsilon_{2}\rangle \right\}$ of two qubits.
(4)$${\hat{\rho }}_{\mathrm{AS},\mathrm{thermal}}=|\Psi^{+}\rangle {\langle \Psi^{+}|}_{\mathrm{AS},\mathrm{thermal}}=\left(\begin{array}{cccc} 0 & 0 & 0 & 0\\ 0 & 1/2 & 1/2 & 0\\ 0 & 1/2 & 1/2 & 0\\ 0 & 0 & 0 & 0 \end{array}\right)$$

This entangled state is designed for d=2 modes, and it corresponds to the shape of entangled states used in Lloyd’s article [4]: ${|\Psi^{+}\rangle }_{\mathrm{AS},\mathrm{thermal}}=1/\sqrt{d}\sum_{i,j=0}^{3}{|\varepsilon_{i}\rangle }_{A}{|\varepsilon_{j}\rangle }_{S}$. In the atmosphere, photon S is submitted to a thermalization by the environment that induces an evolution of its quantum state. An evolution of ${\hat{\rho }}_{S,\mathrm{thermal}}$ implies an evolution of ${\hat{\rho }}_{\mathrm{AS},\mathrm{thermal}}$.

To model such an evolution, we use the article [16], which presents an experiment to study the photon thermalization. This evolution is described by a generalized amplitude-damping channel presented in Section 2.3 that we use to model the evolution of ${\hat{\rho }}_{\mathrm{AS},\mathrm{thermal}}$. The evolution of ${\hat{\rho }}_{\mathrm{AS},\mathrm{thermal}}$ implies the evolution of its quantum information, as we will see in Section 3 and Section 4.

The following section uses Equation (Equation 4) as the input state of the amplitude-damping channel. Note that for this QI radar, we do not consider the reflectivity of the target (η=1), since we work only on the information evolution along the quantum channel modeling the environment.

### 2.3. The Amplitude Damping Channel for the QI Radar

The amplitude-damping quantum channel acts like a heat bath on the qubit S, which propagates in the environment, as seen in step 2b in Figure 1. More precisely, the quantum channel acts locally on the qubit S, but it does not act locally on the qubit A, which is kept in a quantum memory. Therefore, photon A trapped in the radar remains unperturbed by the quantum channel. Moreover, in the QI radar, no local measurement is performed on the subsystem A at any time, as mentioned in Section 2.1. The quantum channel makes the quantum state of Equation (Equation 4) evolve as a function of the parameter *p* described below.

The quantum channel used is taken from the reference [16]. We adapt it to the entangled state of Equation (Equation 4) acting only on the emitted photon S. We drop the subscript “thermal” to simplify the equations; then, we write ${\hat{\rho }}_{\mathrm{AS},\mathrm{thermal}}\equiv {\hat{\rho }}_{\mathrm{AS},\mathrm{in}}$. The amplitude damping channel is defined in Equation (Equation 5).
(5)$$N\left({\hat{\rho }}_{\mathrm{AS},}\;\mathrm{in}\right)=\sum_{i=0}^{3}{\hat{K}}_{i}{\hat{\rho }}_{\mathrm{AS},\mathrm{in}}{\hat{K}}_{i}^{†}={\hat{\rho }}_{\mathrm{AS},\mathrm{out}}$$
where the Kraus operators ${\hat{K}}_{i}$ acting locally on the subsystem S are defined in Equation (6) as functions of the Kraus operators ${\hat{\Gamma }}_{i}$ of the original quantum channel in [16]: ${\hat{K}}_{i}={\hat{I}}_{A}⊗{\hat{\Gamma }}_{i}$, where ${\hat{I}}_{A}$ is the identity matrix on the subsystem A.
(6a)$$\begin{array}{cc} \; & {\hat{K}}_{0}={\hat{I}}_{A}⊗{\hat{\Gamma }}_{0}={\hat{I}}_{A}⊗\sqrt{1-\xi }\left(\begin{array}{cc} 1 & 0\\ 0 & \sqrt{1-p} \end{array}\right) \end{array}$$
(6b)$$\begin{array}{cc} \; & {\hat{K}}_{1}={\hat{I}}_{A}⊗{\hat{\Gamma }}_{1}={\hat{I}}_{A}⊗\sqrt{1-\xi }\left(\begin{array}{cc} 0 & \sqrt{p}\\ 0 & 0 \end{array}\right) \end{array}$$
(6c)$$\begin{array}{cc} \; & {\hat{K}}_{2}={\hat{I}}_{A}⊗{\hat{\Gamma }}_{2}={\hat{I}}_{A}⊗\sqrt{\xi }\left(\begin{array}{cc} 0 & 0\\ \sqrt{p} & 0 \end{array}\right) \end{array}$$
(6d)$$\begin{array}{cc} \; & {\hat{K}}_{3}={\hat{I}}_{A}⊗{\hat{\Gamma }}_{3}={\hat{I}}_{A}⊗\sqrt{\xi }\left(\begin{array}{cc} \sqrt{1-p} & 0\\ 0 & 1 \end{array}\right) \end{array}$$

The Kraus operators verify the unitarity condition $\sum_{i\;=\;0.3}{\hat{K}}_{i}^{†}{\hat{K}}_{i}={\hat{I}}_{\mathrm{AS}}$.

In Equation (6), there are two parameters that control the evolution of ${\hat{\rho }}_{\mathrm{AS},\mathrm{in}}$. The first parameter ξ∈[0,1/2] is the thermal population of the excited state $|\varepsilon_{2}\rangle$. It controls the balance of population on qubit states. For the sake of simplicity, we assume photon S is fully mixed, so we take ξ=1/2 for the computation in Section 3.2 corresponding to our qubit state defined in Equation (Equation 3). However, we keep the general symbol ξ in the equations.

The second parameter $p=1-e^{-\gamma t}\in \left[0,1\right]$ is the damping probability associated with the excited state $|\varepsilon_{2}\rangle$ as a function of the time *t*, where γ is the damping constant. The parameter γ depends on the environment and the frequency photon S. An optical photon is more impacted by the environment than a microwave photon, but setting a value for γ is not within the scope of this paper. γ is only taken as dependent on the environment. Therefore, the important parameter controlling the evolution of the state ${\hat{\rho }}_{\mathrm{AS},\mathrm{in}}$ is p∈[0,1].

Using the quantum channel of Equation (Equation 5) on the density matrix in Equation (Equation 4) gives the output density matrix ${\hat{\rho }}_{\mathrm{AS},\mathrm{out}}$ as a function of *p* and ξ written in Equation (Equation 7).
(7)$${\hat{\rho }}_{\mathrm{AS},\mathrm{out}}=\left(\begin{array}{cccc} \frac{1}{2}p\left(1-\xi \right) & 0 & 0 & 0\\ 0 & \frac{1}{2}\left(1-p+p\xi \right) & \frac{1}{2}\sqrt{1-p} & 0\\ 0 & \frac{1}{2}\sqrt{1-p} & \frac{1}{2}\left(1-p\xi \right) & 0\\ 0 & 0 & 0 & \frac{1}{2}p\xi \end{array}\right)$$

As explained in Section 1, the parameter *p* depends on the propagation environment. It is the main parameter that evolves in our toy model to represent the intensity of the action induced by the propagation environment on the quantum system AS. This density matrix ${\hat{\rho }}_{\mathrm{AS},\mathrm{out}}$ is the starting point for the calculus of the quantum information evolution in Section 3.2. Parameter *p* also permits this quantum information evolution to be linked to the binary decision strategy developed in Section 4. To recap, the parameter has three roles to play. First, it describes the action intensity of the environment on the state described by ${\hat{\rho }}_{\mathrm{AS},\mathrm{out}}$. Second, it binds the model to a practical laboratory case [16]. Third, it binds the amplitude damping channel to Lloyd’s binary decision strategy.

At this point, the amplitude damping channel action on the quantum state AS has been defined in Equation (Equation 7). We will compute the quantum information evolution in the QI radar in the next section.

## 3. Quantum Information in the QI Radar

The first section introduces quantum information theory using classical information theory. Using a particular quantum state, we compare the entanglement and the quantum discord to highlight its difference. In Section 3.2, we compute the quantum information evolution in the quantum channel.

### 3.1. Tools of the Quantum Information Theory

Quantum information theory is an extension of classical information theory for quantum mechanics [18]. The comparison between the mutual information definitions in both theories leads us to the definition of quantum discord. Secondly, we underline the difference between the discord and the entanglement by performing calculations for a Werner state.

In classical information theory, the information about a random variable *X* is defined by Shannon entropy $H_{X}\left(x\right)=-\sum_{x\in X}p_{x}\log\left(p_{x}\right)$. It quantifies the surprise, i.e., the information available for all the realizations *x* of probability $p_{x}$ for the random variable *X*. Considering two random variables *X* and *Y*, we define the mutual information I(X:Y) in Equation (8) in two equivalent formulas, Equation (8a,b). The mutual information I(X:Y) is the common information between these two variables *X* and *Y*.
(8a)$$\begin{array}{cc} I(X:Y) & =H(X)+H(Y)-H(X,Y) \end{array}$$
(8b)$$\begin{array}{cc} I(X:Y) & =H(X)-H(X|Y)=H(Y)-H(Y|X) \end{array}$$

In Equation (8), $H\left(X|Y\right)=-\sum_{x\in X}p_{Y}\left(y\right)H\left(X|y\right)$ is the conditional Shannon entropy of *X*, with knowledge of the realization y∈Y, so it represents the information on *X* conditioned by the realizations *y* on *Y*. Then, we have the joint entropy $H\left(X,Y\right)=-\sum_{x\in X,y\in Y}p_{X,Y}\left(x,y\right)$$\log p_{X,Y}\left(x,y\right)$, where $p_{X,Y}\left(x,y\right)=p_{Y|X}\left(y|x\right)\;p_{X}\left(x\right)$. H(X,Y) is the joint information of the random variables *X* and *Y*. Now, we move to the definition of the mutual information in the quantum information theory.

In quantum information theory, the analog of the Shannon entropy H(X) is the Von Neumann entropy $S\left(\hat{\rho }\right)=-\mathrm{Tr}\left\{\hat{\rho }\;\log\hat{\rho }\right\}$ for a density matrix $\hat{\rho }$. $S(\hat{\rho })$ quantifies the surprise, i.e., the information about a quantum state, since all information about this quantum state is encapsulated in the density matrix $\hat{\rho }$. The sum in classical information theory is replaced by the trace operation Tr{·} in quantum information theory. Similarly, the probability distributions are replaced by the density matrices that describe the quantum states as statistical mixtures of the eigenstates in a Hilbert space. As shown in Equation (8), the quantum mutual information can be defined in two ways in Equation (9); however, neither formula, Equation (9a) nor Equation (9b), is equivalent.
(9a)$$\begin{array}{cc} I({\hat{\rho }}_{\mathrm{AS}}) & =S\left({\hat{\rho }}_{A}\right)+S\left({\hat{\rho }}_{S}\right)-S\left({\hat{\rho }}_{\mathrm{AS}}\right) \end{array}$$
(9b)$$\begin{array}{cc} J{\left({\hat{\rho }}_{\mathrm{AS}}\right)}_{\left\{{\hat{M}}_{S}^{\left(i\right)}\right\}} & =S\left({\hat{\rho }}_{A}\right)-S{\left({\hat{\rho }}_{A}|{\hat{\rho }}_{S}\right)}_{\left\{{\hat{M}}_{S}^{\left(i\right)}\right\}} \end{array}$$

Equation (9a) is the natural extension of the classical mutual information in Equation (8a), while Equation (9b) looks like the conditional information of Equation (8b). Compared to the classical case in Equation (8b), Equation (9b) introduces measurement theory, which is fundamental in quantum mechanics. Consequently, the latter definition reports how much information we can have on A if we measure S.

Equation (9a) is called the “quantum mutual information”, while Equation (9b) is called the “classical information” available in the quantum state ${\hat{\rho }}_{A}$ conditioned by the measurement of the quantum state ${\hat{\rho }}_{S}$. To complete such measurements, we typically use Von Neumann projection operators $\left\{{\hat{M}}_{S}^{\left(i\right)}\right\}=\sum_{i=0}^{1}{\hat{M}}_{S}^{\left(i\right)}$. In Equation (9b), the quantum conditional entropy $S({\hat{\rho }}_{A}|{\hat{\rho }}_{S})$ defined in Equation (Equation 10) depends on the measurement operators.
(10)$$S{\left({\hat{\rho }}_{A}|{\hat{\rho }}_{S}\right)}_{\left\{{\hat{M}}_{S}^{\left(i\right)}\right\}}=\sum_{i=1}^{n}p_{i}S\left({\hat{\rho }}_{A}^{\left(i\right)}\right)$$
where the probability on the state ${\hat{\rho }}_{A}^{\left(i\right)}$ by a measurement on the state “*i*” is $p_{i}=\mathrm{Tr}\left\{\left({\hat{I}}_{A}⊗{\hat{M}}_{S}^{\left(i\right)}\right){\hat{\rho }}_{\mathrm{AS}}\right\}$ and the quantum state ${\hat{\rho }}_{A}^{\left(i\right)}$ is defined by ${\hat{\rho }}_{A}^{\left(i\right)}={\mathrm{Tr}}_{S}\left\{\left({\hat{I}}_{A}⊗{\hat{M}}_{S}^{\left(i\right)}\right){\hat{\rho }}_{\mathrm{AS}}\left({\hat{I}}_{A}⊗{\hat{M}}_{S}^{\left(i\right)}\right)\right\}/p_{i}$.

The quantum discord is then defined in Equation (Equation 11) as the difference between the quantum mutual information $I({\hat{\rho }}_{\mathrm{AS}})$ in Equation (9a) and the classical information $J({\hat{\rho }}_{\mathrm{AS}})$ in Equation (9b) that is accessible in subsystem A by means of a Von Neumann measurement taken of subsystem S.
(11)$$d\left({\hat{\rho }}_{\mathrm{AS}}\right)=\min_{\left\{{\hat{M}}_{S}^{\left(i\right)}\right\}}\left(I\left({\hat{\rho }}_{\mathrm{AS}}\right)-J{\left({\hat{\rho }}_{\mathrm{AS}}\right)}_{\left\{{\hat{M}}_{S}^{\left(i\right)}\right\}}\right)\;$$

This information quantity depends on the measurement operators $\left\{{\hat{M}}_{S}^{\left(i\right)}\right\}$ as it depends on $J({\hat{\rho }}_{\mathrm{AS}})$. Typically, the Von Neumann operators are chosen to verify the minimum required for the quantum discord. Note as that the information $J({\hat{\rho }}_{\mathrm{AS}})$ is not necessarily symmetrical, and the discord $d({\hat{\rho }}_{\mathrm{AS}})$ has the same property. The logarithm functions used to calculate the discord on qubit states are in base 2.

After the quantum discord, we have to define the entanglement rate.

Quantum mechanics only refers to the entanglement of a system, but quantum information theory enables us to quantify the degree of entanglement [17,21]. There are several ways to compute the entanglement rate that depend, for example, on the dimension of the quantum state. As we consider a pair of entangled qubits, we use the concurrence of Wootters to compute the entanglement rate using the entanglement of formation [22].

The entanglement rate $E({\hat{\rho }}_{\mathrm{AS}})$ using the concurrence is calculated using a binary entropy $E\left({\hat{\rho }}_{\mathrm{AS}}\right)\equiv h\left(x\right)=-x\log x-\left(1-x\right)\log\left(1-x\right)$, where $x=(1+\sqrt{1-C^{2}})/2$ with $C=C({\hat{\rho }}_{\mathrm{AS}})$, the concurrence of Wootters [23]. $C({\hat{\rho }}_{\mathrm{AS}})$ is itself an entanglement measure such as C∈[0,1]. If C=1, the state is fully entangled and the entanglement vanishes for C=0. We compute the concurrence $C\left({\hat{\rho }}_{\mathrm{AS}}\right)=\max\left(0,\sqrt{\lambda_{1}}-\sqrt{\lambda_{2}}-\sqrt{\lambda_{3}}-\sqrt{\lambda_{4}}\right)$ with the $\left\{\sqrt{\lambda_{i}}\right\}$, which are the square-root eigenvalues in decreasing order of the spin-flip matrix $R={\hat{\rho }}_{\mathrm{AS}}\left({\hat{\sigma }}_{y}⊗{\hat{\sigma }}_{y}\right){\hat{\rho }}_{\mathrm{AS}}^{*}\left({\hat{\sigma }}_{y}⊗{\hat{\sigma }}_{y}\right)$, where ${\hat{\sigma }}_{y}$ is a Pauli matrix. The symbol * denotes the complex conjugate.

At this point, we define the quantum discord and the entanglement rate. In the following, we highlight the difference between these two quantities using a Werner state (see [15] for details).

To show that both rates $C({\hat{\rho }}_{\mathrm{AS}})$ and $d({\hat{\rho }}_{\mathrm{AS}})$ are different, we take a Werner state ${\hat{\rho }}_{w}=z|\Psi^{-}\rangle {\langle \Psi^{-}|}_{\mathrm{AS}}+\left(1-z\right)/4{\hat{I}}_{\mathrm{AS}}$, which is a mixture of a fully entangled state $|\Psi^{-}\rangle {\langle \Psi^{-}|}_{\mathrm{AS}}$ and a separable state ${\hat{I}}_{\mathrm{AS}}/4$ (= product state) [15]. The parameter z∈[0,1] controls the combination of the two extreme states, so the Werner state is a partially entangled quantum state. In Figure 2, we plot the concurrence, the entanglement rate and the quantum discord as functions of parameter z. The three quantities decrease as z tends toward zero.

We focus our attention on the rate $E({\hat{\rho }}_{\mathrm{AS}})$ and $d({\hat{\rho }}_{\mathrm{AS}})$. In Figure 2, the entanglement rate and the quantum discord are maximal for z=1, which corresponds to a fully entangled state $|\Psi^{-}\rangle {\langle \Psi^{-}|}_{\mathrm{AS}}$. $E({\hat{\rho }}_{\mathrm{AS}})=1$ means the quantum state is fully entangled, and $d({\hat{\rho }}_{\mathrm{AS}})=1$ means we have reached the maximum number of possible quantum correlations.

The entanglement rate and the discord are zero when z=0, corresponding to the fully mixed state ${\hat{I}}_{\mathrm{AS}}/4$. $E({\hat{\rho }}_{\mathrm{AS}})=0$ means that the state is no longer entangled, and we also have $d({\hat{\rho }}_{\mathrm{AS}})=0$, so all quantum correlations have vanished.

Now, looking at the intermediate z∈]0,1[, we can see that both rates take different values. The entanglement rate decay is faster than the discord rate as z tends towards zero. For $z_{d}=1/3$, we observe the entanglement rate $E({\hat{\rho }}_{w})=0$, the concurrence $C({\hat{\rho }}_{w})=0$ contrary to the quantum discord $d({\hat{\rho }}_{w})>0$. This means the state ${\hat{\rho }}_{w}$ for $z_{d}=1/3$ is no longer entangled, but it still has quantum correlations. These correlations are not due to the entanglement as the state is separable; i.e., the state is not entangled. Figure 2 shows that entanglement and discord translate to two different kinds of quantum correlations, so there are quantum correlations beyond the entanglement.

It ought to be noted that we can have a quantum state that is not entangled with a non-zero discord, but the contrary is not possible. We cannot have an entangled state with a null discord. It is recalled that an entangled state cannot be written as a product state, called a separable state. Looking at Figure 2, when $z_{d}=1/3$, we can see that the state becomes separable, but it has a non-zero discord for z∈[0,1/3]. Then, the entanglement is a quantum phenomenon, while the quantum discord is a quantum correlation measurement. The entanglement rate $E({\hat{\rho }}_{w})$ does not depend on projective measurements such as the discord $d({\hat{\rho }}_{w})$. For the latter, we have to be particularly attentive when selecting the measurement operators to verify Equation (Equation 11).

We have now defined the difference between the entanglement rate and the quantum discord. The next section uses the tools of information theory described here to compute the evolution of the quantum information along the amplitude damping quantum channel defined in Section 2.3.

### 3.2. Evolution of the Quantum Information in the QI Radar

In this section, we compute the entanglement rate and the quantum discord in the amplitude-damping quantum channel of Equation (Equation 5) as functions of the damping parameter *p*. We begin by calculating the entanglement rate before calculating the quantum discord.

The entanglement rate is calculated using the concurrence of Wootters $C({\hat{\rho }}_{\mathrm{AS},\mathrm{out}})$, defined in Section 3.1. We compute the matrix $R={\hat{\rho }}_{\mathrm{AS},\mathrm{out}}\left({\hat{\sigma }}_{y}⊗{\hat{\sigma }}_{y}\right){\hat{\rho }}_{\mathrm{AS},\mathrm{out}}^{*}\left({\hat{\sigma }}_{y}⊗{\hat{\sigma }}_{y}\right)$, and we extract its eigenvalues. In Equation (12), we write the square-roots of the eigenvalues $\sqrt{\lambda_{i}}$ of the matrix *R*.
(12a)$$\begin{array}{cc} \; & \sqrt{\lambda_{1}}=\frac{1}{2}p\sqrt{\xi (1-\xi )} \end{array}$$
(12b)$$\begin{array}{cc} \; & \sqrt{\lambda_{2}}=\sqrt{\frac{1}{2}\left[\frac{1}{2}\left(1-p+\left(1-p+p\xi \right)\left(1-p\xi \right)\right)+\sqrt{\Delta (R)}\right]} \end{array}$$
(12c)$$\begin{array}{cc} \; & \sqrt{\lambda_{3}}=\sqrt{\frac{1}{2}\left[\frac{1}{2}\left(1-p+\left(1-p+p\xi \right)\left(1-p\xi \right)\right)-\sqrt{\Delta (R)}\right]} \end{array}$$
(12d)$$\begin{array}{cc} \; & \Delta (R)=(1-p)(1-p\xi )(1-p+p\xi ) \end{array}$$

The eigenvalue $\sqrt{\lambda_{1}}$ has a multiplicity of two, while $\sqrt{\lambda_{2}}$ and $\sqrt{\lambda_{3}}$ have a multiplicity of one. For each value of *p*, these eigenvalues are sorted in decreasing order to compute the concurrence $C({\hat{\rho }}_{\mathrm{AS},\mathrm{out}})$ and next to evaluate the entanglement rate $E({\hat{\rho }}_{\mathrm{AS},\mathrm{out}})$. Both rates are displayed in Figure 3, where we observe that the entanglement is lost before the parameter *p* reaches its limiting value 1. Indeed, we find $C({\hat{\rho }}_{\mathrm{AS},\mathrm{out}})=0$ for p≈0.83, so for p>0.83, the system AS is no longer entangled. Therefore, the fully entangled state of Equation (Equation 7) becomes a separable state before the limiting value p=1.

Now, we will evaluate the quantum discord $d({\hat{\rho }}_{\mathrm{AS},\mathrm{out}})$ by calculating, in this order, the Von Neumann entropies $S({\hat{\rho }}_{A,\mathrm{out}})$, $S({\hat{\rho }}_{S,\mathrm{out}})$, $S({\hat{\rho }}_{\mathrm{AS},\mathrm{out}})$, and $S({\hat{\rho }}_{A,\mathrm{out}}|{\hat{\rho }}_{S,\mathrm{out}})$ as functions of *p*. With these entropies, we compute the quantum mutual information $I({\hat{\rho }}_{\mathrm{AS},\mathrm{out}})$ and the classical information $J({\hat{\rho }}_{\mathrm{AS},\mathrm{out}})$ from Equation (9) to estimate the quantum discord using Equation (Equation 11).

Using Equation (Equation 7), we calculate the density matrices of both subsystems A and S using a partial trace operation: ${\hat{\rho }}_{A,\mathrm{out}}={\mathrm{Tr}}_{S}\left\{{\hat{\rho }}_{\mathrm{AS},\mathrm{out}}\right\}$ and ${\hat{\rho }}_{S,\mathrm{out}}={\mathrm{Tr}}_{A}\left\{{\hat{\rho }}_{\mathrm{AS},\mathrm{out}}\right\}$. Next, we calculate the Von Neumann entropies of both subsystems in Equation (13).
(13a)$$\begin{array}{cc} S({\hat{\rho }}_{A,\mathrm{out}}) & =1\;\forall p\in [0,1] \end{array}$$
(13b)$$\begin{array}{c} \begin{array}{cc} S({\hat{\rho }}_{S,\mathrm{out}}) & =\frac{1+p-2p\xi }{2}\log\left(\frac{2}{1+p-2p\xi }\right)+\frac{1-p+2p\xi }{2}\log\left(\frac{2}{1-p+2p\xi }\right) \end{array} \end{array}$$

Note that as we set ξ=1/2 in Section 2.3, we have $S\left({\hat{\rho }}_{A,\mathrm{out}}\right)=S\left({\hat{\rho }}_{S,\mathrm{out}}\right)=1$∀p∈[0,1].

According to Equation (13a,b), the information in subsystems A and S is not affected by the action of the quantum channel. We have exactly the same amount of information for all values of parameter p∈[0,1] for photon A and for photon S.

Now, we apply the same calculation for the entropy $S\left({\hat{\rho }}_{\mathrm{AS},\mathrm{out}}\right)=-\mathrm{Tr}\left\{{\hat{\rho }}_{\mathrm{AS},\mathrm{out}}\log{\hat{\rho }}_{\mathrm{AS},\mathrm{out}}\right\}$ on the entangled system AS. It should be noted that the logarithm used is in base 4, log(x)=ln(x)/ln(4) as $\dim({\hat{\rho }}_{\mathrm{AS},\mathrm{out}})=4$.

The matrix logarithm $\log({\hat{\rho }}_{\mathrm{AS}})$ can be calculated analytically by diagonalizing the density matrix ${\hat{\rho }}_{\mathrm{AS},\mathrm{out}}$ to obtain $\log\left({\hat{\rho }}_{\mathrm{AS},\mathrm{out}}\right)=P\;\log\left(D\right)\;P^{-}1$, where *P* is the modal matrix and $P^{-1}$ is its inverse. *D* is the diagonal matrix formed by the eigenvalues of ${\hat{\rho }}_{\mathrm{AS},\mathrm{out}}$. The calculation is detailed in Appendix A within which we wrote out the Von Neumann entropy $S\left({\hat{\rho }}_{\mathrm{AS},\mathrm{out}}\right)=-H_{1}-H_{2}-H_{5}-H_{6}$ with coefficients $\left\{H_{i=1\ldots 6}\right\}$, also defined as functions of *p* in Appendix A. The evolution of $S({\hat{\rho }}_{\mathrm{AS},\mathrm{out}})$ is plotted in Figure 4. We observe that the Von Neumann increases as *p* tends towards one as we lose the quantum correlations because of the quantum channel action. For p=0, $S({\hat{\rho }}_{\mathrm{AS},\mathrm{out}})=0$ corresponding to the fully entangled pure state and next the entropy increases because the uncertainty on the state ${\hat{\rho }}_{\mathrm{AS},\mathrm{out}}$ increases. Indeed, as *p* tends towards one, we progress to a fully mixed separable state.

Here, we compute the conditional entropy $S{\left({\hat{\rho }}_{A,\mathrm{out}}|{\hat{\rho }}_{S,\mathrm{out}}\right)}_{\left\{{\hat{M}}_{S}^{\left(i\right)}\right\}}$ using the projection operators $\left\{{\hat{M}}_{S}^{\left(i=1,2\right)}\right\}$ with ${\hat{M}}_{S}^{\left(1\right)}={\hat{I}}_{A}⊗|\varepsilon_{1}\rangle {\langle \varepsilon_{1}|}_{S}$ and ${\hat{M}}_{S}^{\left(2\right)}={\hat{I}}_{A}⊗|\varepsilon_{2}\rangle {\langle \varepsilon_{2}|}_{S}$. We obtain the conditional entropy in Equation (Equation 14) that is plotted in Figure 4.
(14)$$\begin{array}{c} \begin{array}{cc} S{\left({\hat{\rho }}_{A,\mathrm{out}}|{\hat{\rho }}_{S,\mathrm{out}}\right)}_{\left\{{\hat{M}}_{S}^{\left(i\right)}\right\}} & =\frac{p(1-\xi )}{2}\log\left(\frac{1+p-2p\xi }{p(1-\xi )}\right)+\frac{1-p\xi }{2}\log\left(\frac{1+p-2p\xi }{1-p\xi }\right)\\ \; & +\frac{1-p+p\xi }{2}\log\left(\frac{1-p+2p\xi }{1-p+p\xi }\right)+\frac{p\xi }{2}\log\left(\frac{1-p+2p\xi }{p\xi }\right) \end{array} \end{array}$$

The conditional entropy increases as *p* tends towards one, and we observe its increase as the quantum state ${\hat{\rho }}_{\mathrm{AS},\mathrm{out}}$ loses its entanglement in Figure 3. The entanglement loss means we progressively lose the quantum correlations between the quantum state ${\hat{\rho }}_{A,\mathrm{out}}$ and the quantum state ${\hat{\rho }}_{S,\mathrm{out}}$. Consequently, we increase the uncertainty between both states A and S when a projective measurement is completed for ${\hat{\rho }}_{S,\mathrm{out}}$.

Using the calculations of the entropies, we can now compute the quantum mutual information $I({\hat{\rho }}_{\mathrm{AS},\mathrm{out}})$ and the classical information $J({\hat{\rho }}_{\mathrm{AS},\mathrm{out}})$ as functions of the parameter *p* using Equation (9) to plot them in Figure 5. The quantum mutual information is maximal for the fully entangled state when p=0, and it decreases to zero until p=1. We observe the same evolution for the classical information $J({\hat{\rho }}_{\mathrm{AS},\mathrm{out}})$. The quantum mutual information $I({\hat{\rho }}_{\mathrm{AS},\mathrm{out}})$ represents the total information shared by both subsystems A and S. Its decreasing evolution in Figure 5 is in line with the entanglement decay observed in Figure 3. The entanglement decay means the destruction of the initial quantum information shared by both subsystems because the quantum correlations are vanishing. The classical information decay $J({\hat{\rho }}_{\mathrm{AS},\mathrm{out}})$ shows that we lose the ability to extract information from state ${\hat{\rho }}_{A,\mathrm{out}}$ when projective measurements are taken in state ${\hat{\rho }}_{S,\mathrm{out}}$. A comparison between Figure 3 and Figure 5 shows that when the entanglement vanishes at p≈0.83, the classical information $J({\hat{\rho }}_{\mathrm{AS},\mathrm{out}})$ is close to zero but not the quantum mutual information $I({\hat{\rho }}_{\mathrm{AS},\mathrm{out}})$.

Finally, we compute the quantum discord $d({\hat{\rho }}_{\mathrm{AS},\mathrm{out}})$ as a function of parameter *p* in Figure 6, including the entanglement rates $C({\hat{\rho }}_{\mathrm{AS},\mathrm{out}})$ and $E({\hat{\rho }}_{\mathrm{AS},\mathrm{out}})$. We clearly see that the entanglement rate of decay is faster than the quantum discord decay as *p* tends towards one.

When p>0.83, the entanglement rate vanishes but not the quantum discord. This means that some quantum correlations survived to the amplitude damping quantum channel in Equation (Equation 5) despite the state ${\hat{\rho }}_{\mathrm{AS},\mathrm{out}}{|}_{p>0.83}$ no longer being entangled, i.e., it is a separable state. When *p* tends towards one, the quantum discord vanishes as well, so all quantum correlations are lost and we obtain a maximally mixed separable state that is a product state ${\hat{\rho }}_{A,\mathrm{out}}⊗{\hat{\rho }}_{S,\mathrm{out}}=\frac{1}{4}{\hat{I}}_{A}⊗{\hat{I}}_{S}$.

Through calculations of the entanglement rate and the quantum discord in this section, we observed the presence of quantum correlations surviving slightly after the entanglement loss. In Lloyd’s article [4], we keep the quantum advantage in terms of sensitivity even when the entanglement is lost. Reasoning with a quantum information approach allows us to understand that it is the existence of quantum information, i.e., quantum correlations, that produce the sensitivity enhancement. However, keep in mind that quantum correlations will certainly vanish in atmosphere after a finite propagation time, which leads to the question of the quantum radar ranging. Indeed, if entanglement is lost but not the quantum advantage, it is when the quantum correlations are totally lost that our sensitivity enhancement would collapse. Hence, even if the model presented is not realistic, it is worth linking the quantum channel action with the decision strategy presented in Section 4, since the entanglement rate and the quantum discord do not evolve at the same speed as function of the damping parameter *p*. This comparison between the amplitude damping channel and the binary decision strategy permit us to combine the propagation phase with the detection phase including thermal noise.

The next section presents Lloyd’s decision strategy, accounting for the quantum channel defined in Section 2.3. Section 4 shows the link between the quantum information evolution in the amplitude-damping quantum channel and Lloyd’s binary decision strategy for the QI radar [4].

## 4. The Binary Decision Strategy for the QI Radar

In this section, we introduce Lloyd’s binary decision strategy for the QI radar, establishing the link with the amplitude damping quantum channel of Section 2.3.

We start by explaining Lloyd’s original decision strategy, and we then adapt this strategy according to the quantum channel action to calculate the signal-to-noise ratios (SNR) depending on the damping probability *p*.

In Lloyd’s article [4], the binary decision strategy used relies on the discrimination of quantum states [24]. This discrimination of quantum states is enhanced by the entanglement in a QI radar compared to a single-photon radar. The binary decision can only provide the information on the absence or the presence of an object with a reflectivity 0⩽η⩽1 surrounded by a thermal noise. These hypotheses are, respectively, called hypothesis $H_{0}$ and hypothesis $H_{1}$. Figure 7 provides an illustration of the situation depicted by both hypotheses. At the top of Figure 7, we have hypothesis $H_{1}$, where a target can be detected. On the bottom, we have hypothesis $H_{0}$, where only thermal noise photons are present. The construction of the approximations for the thermal quantum states associated with each hypothesis is possible for two reasons. The first reason is that the average number of thermal photons $n_{b}=\left(1-e^{-\beta \hbar \omega }\right)e^{-\beta \hbar \omega }$ per thermal energy mode is very low: $n_{b}\ll 1$. This average number $n_{b}$ is calculated from Planck’s law $n\left(\omega ,T\right)={\left(e^{\beta \hbar \omega }-1\right)}^{-1}$. This approximation is suitable for the optical frequency regime but not for the microwave frequency regime. The second reason is that the photodetector can distinguish $d=W\times t_{d}$ modes per detection event, as asserted in Section 2.1. Moreover, the photodetector can detect one photon at most per detection event, corresponding to a number of thermal photons detected $dn_{b}\ll 1$.

For Lloyd’s QI radar, the detection works on a reflecting target with a reflectivity η, and the assumption was that the environment destroys the entanglement instantly, but the sensitivity enhancement is maintained.

In this paper, our toy model considers Lloyd’s QI radar, but we only consider the influence of the propagation environment as we do not account for the target. Therefore, we consider the target as a perfectly reflecting object (η=1) that reflects the incident photon S towards our quantum radar with certainty. Such a simplification is obviously unrealistic since the photon can be reflected anywhere in practice. We do assume that the photon comes back to the radar in the toy model. Thus, strictly speaking, we do not perform a true QI radar detection under such simplifications. Our objective is to study the evolution of quantum information in the damping channel with the signal-to-noise ratios (SNR) calculated with the binary decision strategy.

The assumptions for calculations are the same as those in the previous paragraph, but we limit ourselves to the case d=2 modes, and hence the number of thermal photons seen by the photodetector is $2n_{b}\ll 1$, which is obviously not realistic. Nevertheless, it permits us to work with thermal qubits in the decision theory and to use the amplitude damping channel acting as a heat bath on a qubit state as stated in [16]. The link with the channel is defined by the damping probability $p=1-e^{-\gamma t}$ described in Section 2.3. This parameter p∈[0,1] shows that the quantum channel action increases as *p* tends towards one. Next, we will take Lloyd’s binary decision strategy and adapt it to our model.

In light of the reference article [4], we consider a single-photon radar and a quantum illumination radar. We begin by defining the thermal states on the single-photon radar for hypotheses $H_{0}$ and $H_{1}$. Next, we use these thermal states to define the states on the pair of entangled qubits following the QI radar depicted in Figure 1 for both hypotheses $H_{0}$ and $H_{1}$. In both radar scenarios, we compute the probability of detection error for single-shot measurements.

For the single-photon radar, according to Lloyd [4], the approximation of the thermal state found for hypothesis $H_{0}$ is given by Equation (Equation 15)
(15)$${\hat{\rho }}_{0}=\left(1-2n_{b}\right)|vac\rangle \langle vac|+n_{b}\sum_{i=1}^{2}|\varepsilon_{i}\rangle \langle \varepsilon_{i}|+O\left({\left(2n_{b}\right)}^{2}\right)$$
where $|vac\rangle$ is the vacuum state on the thermal energy modes and $|\varepsilon_{i}\rangle$ are the thermal modes populated by one photon. The photodetector cannot have a useful signal, since there is no object to reflect the photon. The photodetector can only see the vacuum or a thermal photon over the d=2 modes.

In hypothesis $H_{1}$, the object is present, and it can reflect the emitted photon. We obtain Equation (Equation 16), whereby we have a probability *p* of having only the thermal state of Equation (Equation 15) given the quantum channel influence, or we can retrieve photon S thermalized by the propagation environment.
(16)$$\begin{array}{c} \begin{array}{cc} {\hat{\rho }}_{1} & =p{\hat{\rho }}_{0}+\left(1-p\right){\hat{\rho }}_{S}\\ \; & =p\left[\left(1-2n_{b}\right)|vac\rangle \langle vac|+n_{b}\sum_{i=1}^{2}|\varepsilon_{i}\rangle \langle \varepsilon_{i}|\right]+\left(1-p\right)|\varphi \rangle {\langle \varphi |}_{S}+O\left({\left(2n_{b}\right)}^{2}\right) \end{array} \end{array}$$
where ${|\varphi \rangle }_{S}=1/\sqrt{2}\left(|\varepsilon_{1}\rangle +|\varepsilon_{2}\rangle \right)$ is the qubit state of photon S.

From this point, we have to define the probability of detection error for single-shot measurements. We use the reference [24] and Lloyd’s article [4]. It consists in completing projective measurements on the positive and the negative parts of the operator $\left({\hat{\rho }}_{1}-{\hat{\rho }}_{0}\right)$ in Equation (Equation 17).
(17)$$\begin{array}{c} \begin{array}{cc} \left({\hat{\rho }}_{1}-{\hat{\rho }}_{0}\right) & =p{\hat{\rho }}_{0}+\left(1-p\right){\hat{\rho }}_{S}-{\hat{\rho }}_{0}\\ \; & =p\left[\left(1-2n_{b}\right)|vac\rangle \langle vac|+n_{b}\sum_{i=1}^{2}|\varepsilon_{i}\rangle \langle \varepsilon_{i}|\right]+\left(1-p\right)|\varphi \rangle {\langle \varphi |}_{S}\\ \; & -\left[\left(1-2n_{b}\right)|vac\rangle \langle vac|+n_{b}\sum_{i=1}^{2}|\varepsilon_{i}\rangle \langle \varepsilon_{i}|\right] \end{array} \end{array}$$

The negative part corresponds to hypothesis $H_{0}$, and the positive part corresponds to hypothesis $H_{1}$. Therefore, the probability to detect one particular state is a conditional probability to obtain a positive/negative result, knowing the starting hypothesis $H_{1}$ or $H_{0}$. The probabilities of detection error are conditional probabilities and are written in Table 1.

Following the QI radar scheme depicted in Figure 1, we take into account emitted photon S as well as ancilla photon A. We completed the same calculations for the pair of entangled qubits starting with hypothesis $H_{0}$, whereby we obtained only a separable thermal state, since there is no object to detect. This quantum state is written in Equation (Equation 18) according to the reference [4].
(18)$${\hat{\rho }}_{\mathrm{AS}}^{\left(H_{0}\right)}=\frac{{\hat{I}}_{A}}{2}⊗{\hat{\rho }}_{0}=\frac{{\hat{I}}_{A}}{2}⊗\left[\left(1-2n_{b}\right)|vac\rangle \langle vac|+n_{b}\sum_{i=1}^{2}|\varepsilon_{i}\rangle \langle \varepsilon_{i}|\right]$$

In Equation (Equation 18), we can detect a thermal noise photon over the thermal energy modes $|\varepsilon_{i}\rangle$ corresponding to a product state such as ${\hat{I}}_{A}⊗{\hat{I}}_{S}$, while the alternative possibility is to detect only the vacuum $|vac\rangle$.

For hypothesis $H_{1}$, we obtain Equation (Equation 19), whereby we have the probability *p* to obtain the thermal state of Equation (Equation 18) because of the quantum channel influence, or we can retrieve the entangled state $|\Psi \rangle {\langle \Psi |}_{\mathrm{AS}}$.
(19)$$\begin{array}{c} \begin{array}{cc} {\hat{\rho }}_{\mathrm{AS}}^{\left(H_{1}\right)} & =p\;\frac{{\hat{I}}_{A}}{2}⊗{\hat{\rho }}_{0}+\left(1-p\right)|\Psi \rangle {\langle \Psi |}_{\mathrm{AS}}\\ \; & =p\frac{{\hat{I}}_{A}}{2}⊗\left[\left(1-2n_{b}\right)|vac\rangle \langle vac|+n_{b}\sum_{i=1}^{2}|\varepsilon_{i}\rangle \langle \varepsilon_{i}|\right]+\left(1-p\right)|\Psi \rangle {\langle \Psi |}_{\mathrm{AS}} \end{array} \end{array}$$

In Equation (Equation 19), we have a decreasing probability (1−p) to retrieve a useful signal as *p* tends towards one. The probabilities of detection error for single-shot measurements are calculated in Equation (Equation 20) with the operator $\left({\hat{\rho }}_{\mathrm{AS}}^{\left(H_{1}\right)}-{\hat{\rho }}_{\mathrm{AS}}^{\left(H_{0}\right)}\right)$ similarly to the single- photon radar.
(20)$$\begin{array}{c} \begin{array}{cc} \left({\hat{\rho }}_{\mathrm{AS}}^{\left(H_{1}\right)}-{\hat{\rho }}_{\mathrm{AS}}^{\left(H_{0}\right)}\right) & =p\;\frac{{\hat{I}}_{A}}{2}⊗{\hat{\rho }}_{0}+\left(1-p\right)|\Psi \rangle {\langle \Psi |}_{\mathrm{AS}}-\frac{{\hat{I}}_{A}}{2}⊗{\hat{\rho }}_{0}\\ \; & =p\;\frac{{\hat{I}}_{A}}{2}⊗\left[\left(1-2n_{b}\right)|vac\rangle \langle vac|+n_{b}\sum_{i=1}^{2}|\varepsilon_{i}\rangle \langle \varepsilon_{i}|\right]+\left(1-p\right)|\Psi \rangle {\langle \Psi |}_{\mathrm{AS}}\\ \; & -\frac{{\hat{I}}_{A}}{2}⊗\left[\left(1-2n_{b}\right)|vac\rangle \langle vac|+n_{b}\sum_{i=1}^{2}|\varepsilon_{i}\rangle \langle \varepsilon_{i}|\right] \end{array} \end{array}$$

The positive part represents hypothesis $H_{1}$ and the negative part represents hypothesis $H_{0}$. The probabilities of detection error are represented by the conditional probabilities to obtain a negative or a positive result given the starting hypothesis in Table 1.

In Table 1, logically and in line with the observations of Lloyd, we observe that the probabilities of detection error are enhanced by the number *d* of modes involved in the pair of entangled qubits. Of course, our toy model gives only an enhancement by 1/2, which is the limiting case. Below, we calculate the SNR for the single-photon radar and for the QI radar in each hypothesis starting with the single-photon radar.

For the single-photon radar, the signal-to-noise ratios (SNRs) for hypotheses $H_{0}$ and $H_{1}$ consist in calculating the ratio of $P_{e}\left(+\right)/P_{e}\left(-\right)$ to obtain the SNR written in Equation (21).
(21a)$$\begin{array}{cc} \; & {\mathrm{SNR}}_{H_{0}}=\frac{P_{e}\left(+|H_{0}\right)}{P_{e}\left(-|H_{0}\right)}=\frac{n_{b}}{1-n_{b}} \end{array}$$
(21b)$$\begin{array}{cc} \; & {\mathrm{SNR}}_{H_{1}}=\frac{P_{e}\left(+|H_{1}\right)}{P_{e}\left(-|H_{1}\right)}=\frac{1-p+pn_{b}}{n_{b}} \end{array}$$
(21c)$$\begin{array}{cc} \; & {\mathrm{SNR}}_{+}=\frac{P_{e}\left(+|H_{1}\right)}{P_{e}\left(+|H_{0}\right)}=\frac{1-p+pn_{b}}{p(1-n_{b})} \end{array}$$

In Equation (21), all the SNR depend on the damping parameter *p* except for the ${\mathrm{SNR}}_{H_{0}}$. Note when $p=1-e^{-\gamma t}$, with a constant γ, tends towards one it corresponds to an infinite propagation, so we are sure to lose the emitted qubit or the quantum correlations in the QI radar. Looking at Equation (21) and recalling that $n_{b}\ll 1$, we see that the ${\mathrm{SNR}}_{H_{0}}$ depends exclusively on the thermal noise $n_{b}$. We can associate the ${\mathrm{SNR}}_{H_{0}}$ with the probability of a false alarm to detect a signal when there is no object. As $n_{b}\ll 1$, we have ${\mathrm{SNR}}_{H_{0}}<1$, so the probability of false alarm is low. For ${\mathrm{SNR}}_{H_{1}}$, we have $pn_{b}\ll 1$, since p∈[0,1] and $n_{b}\ll 1$, so for a low value of *p*, we obtain ${\mathrm{SNR}}_{H_{1}}>1$. However, when p→1, the ${\mathrm{SNR}}_{H_{1}}$ tends towards zero, so the quantum channel action makes the SNR collapse. For the ${\mathrm{SNR}}_{+}$, we compare the useful signal of hypothesis $H_{1}$ to the noise signal of hypothesis $H_{0}$. As $n_{b}\ll 1$ and p→1, we obtain a ${\mathrm{SNR}}_{+}\approx \left(1-p\right)/p$, so the SNR decreases quickly to zero because of the factor (1−p). However, for small values of *p*, the ${\mathrm{SNR}}_{+}$ is greater than one, because the quantum channel action is not strong enough.

To compare the detection efficiency between both radars, we look at the ${\mathrm{SNR}}_{+}$, since it takes into account both hypotheses. The quantum channel interacts with the emitted photon, producing the SNR decay until it vanishes for p=1.

In the QI radar, the signal-to-noise ratios in Equation (22) are computed for hypotheses $H_{0}$ and $H_{1}$ by calculating the ratio of $P_{e}\left(+\right)/P_{e}\left(-\right)$ as in the single-photon radar.
(22a)$$\begin{array}{cc} \; & {\mathrm{SNR}}_{H_{0}}=\frac{P_{e}\left(+|H_{0}\right)}{P_{e}\left(-|H_{0}\right)}=\frac{n_{b}/2}{1-n_{b}/2} \end{array}$$
(22b)$$\begin{array}{cc} \; & {\mathrm{SNR}}_{H_{1}}=\frac{P_{e}\left(+|H_{1}\right)}{P_{e}\left(-|H_{1}\right)}=\frac{1-p+pn_{b}/2}{p(1-n_{b}/2)} \end{array}$$
(22c)$$\begin{array}{cc} \; & {\mathrm{SNR}}_{+}=\frac{P_{e}\left(+|H_{1}\right)}{P_{e}\left(+|H_{0}\right)}=\frac{1-p+pn_{b}/2}{p(1-n_{b}/2)} \end{array}$$

As shown by Seth Lloyd in their article [4], the SNR calculated from Table 1 benefits from the number of entangled modes. For instance, the ${\mathrm{SNR}}_{H_{0}}$ interpreted as the probability of false alarm is lower than in the ${\mathrm{SNR}}_{H_{0}}$ of the single-photon radar thanks to the factor of 1/2 originating from the entangled modes. For the ${\mathrm{SNR}}_{H_{1}}$, as we have $n_{b}/2\ll 1$, ${\mathrm{SNR}}_{H_{1}}\approx \left(1-p\right)/p$ so the ${\mathrm{SNR}}_{H_{1}}$ is greater than one for small values of *p*. However, it vanishes as *p* tends towards one because of the thermalization induced by the quantum channel. For the ${\mathrm{SNR}}_{+}$, as $n_{b}/2\ll 1$, we also obtain ${\mathrm{SNR}}_{+}\approx \left(1-p\right)/p$, so the SNR is greater than one for low values of *p* but vanishes as *p* tends towards one. The ${\mathrm{SNR}}_{+}$ shows the comparison between hypotheses $H_{0}$ and $H_{1}$, and as for the single-photon radar, we lose the ${\mathrm{SNR}}_{+}$ for p=1. We plotted the ${\mathrm{SNR}}_{+}$ of QI radar in Figure 8, which is linear with the damping probability *p*.

The ${\mathrm{SNR}}_{H_{1}}$ and the ${\mathrm{SNR}}_{+}$ in the single-photon radar and in the QI radar have the same behavior. The SNR is greater than one when *p* is low, but it finally vanishes when *p* tends toward one. The link between both radar situations is the amplitude damping channel acting on the emitted qubit, i.e., the qubit S in Figure 1. The quantum channel thermalizes the qubit S, which propagates, and the longer the propagation is, the greater the perturbation on the qubit state is. More precisely, the closer to one the parameter *p* is, the longer the propagation is. In both radars, the quantum channel perturbs the emitted photon, but, the most interesting observation is of the effect on the QI radar, as for *p* close to one, we have a greater probability to obtain a separable thermal state ${\hat{I}}_{A}/2⊗{\hat{\rho }}_{0}$, as written in Equation (Equation 19).

We recall the different results in the damping channel model and in the binary decision strategy to clarify the link between both models.

The generalized amplitude-damping channel used from the article [16] permits us to describe the evolution of the entangled quantum state ${\hat{\rho }}_{\mathrm{AS},\mathrm{out}}$. This channel models the decoherence of the maximally entangled quantum state as a function of the damping parameter $p=1-e^{-\gamma t}$, where γ depends on the atmosphere. Although the value γ is qualitative, we can safely say that the decoherence is greater in the optical domain compared to the microwave domain. In this paper, we stay in the optical domain to verify the assumptions in the binary decision strategy made in Section 4. At the input of the channel, we use the fully entangled state $|\Psi^{+}\rangle {\langle \Psi^{+}|}_{\mathrm{AS}}$, and we obtain at the output a fully mixed separable state ${\hat{I}}_{A}/2⊗{\hat{I}}_{S}/2$. With the channel, we have the environmental action.

In Lloyd’s binary decision strategy, we consider only the amount of noise for the photodetection to obtain the SNR according to the hypotheses $H_{0}$ and $H_{1}$. In Equation (22), we have the ${\mathrm{SNR}}_{+}$, which compares hypothesis $H_{1}$ with hypothesis $H_{0}$. This comparison between the hypotheses gives us the ratio between the useful signal and the noise from the environment. Looking at Figure 8 shows that the evolution of ${\mathrm{SNR}}_{+}$ is linear. We have the maximum ${\mathrm{SNR}}_{+}=1$ for p=0, while the ${\mathrm{SNR}}_{+}$ vanishes for p=1. Please note that the calculated ${\mathrm{SNR}}_{+}$ has been normalized because the thermal noise in the optical domain is very low. Furthermore, in Equation (Equation 20), the maximum of ${\mathrm{SNR}}_{+}$ corresponds to the fully entangled state $|\Psi \rangle {\langle \Psi |}_{\mathrm{AS}}$, while SNR=0 corresponds to the fully mixed state ${\hat{I}}_{A}/2⊗n_{b}\sum_{i=1}^{2}|\varepsilon_{i}\rangle \langle \varepsilon_{i}|$. Lloyd’s binary decision strategy has been elaborated with the damping probability *p* used in the channel (Equation (Equation 5)). Therefore, both models are linked by this quantity, which allows us to compare the states described by ${\hat{\rho }}_{\mathrm{AS},\mathrm{out}}$ and $|\Psi \rangle {\langle \Psi |}_{\mathrm{AS}}$.

Recall that an open quantum system is a system traveling into a perturbative environment that produces its evolution towards a fully mixed state [25]. Looking at Figure 6, we see that the state ${\hat{\rho }}_{\mathrm{AS},\mathrm{out}}$ is no longer entangled in the interval p∈[0.83,1], but the quantum discord is different from zero. Hence, on this interval, the quantum state ${\hat{\rho }}_{\mathrm{AS},\mathrm{out}}$ is separable, but it still has quantum correlations. Now, as shown in Figure 8, the signal-to-noise ratio does not vanish: we have the $0<{\mathrm{SNR}}_{+}<1$ on the interval p∈[0.83,1]. The ${\mathrm{SNR}}_{+}$ is not zero when the state is no longer entangled, but it still has some quantum correlations. This shows the role of quantum discord in the binary decision strategy for the QI radar.

The damping channel and the decision strategy are coupled by the parameter *p*, so the extremum states coincide. However, we do not have a perfect correspondence for the separable state, since in the damping channel (Equation (Equation 5)), we obtain a fully mixed state ${\hat{I}}_{\mathrm{AS}}/4$, and we obtain the fully mixed state ${\hat{I}}_{A}/2⊗n_{b}\sum_{i=1}^{2}|\varepsilon_{i}\rangle \langle \varepsilon_{i}|$ for the decision strategy. The only difference comes from the noise $n_{b}$ introduced in the binary decision strategy, which does not appear explicitly in the amplitude-damping channel. In spite of this difference, the separable states obtained are similar.

Finally, when the discord becomes zero, this means that the quantum correlations vanish. We have the maximally mixed separable state in both models. The ${\mathrm{SNR}}_{+}=0$ for this state because we cannot make a proper discrimination between the thermal qubit sent in the atmosphere and one thermal qubit from the thermal noise. We can see the vanishing discord as the operational limit scale of the QI radar.

Specifically, the parallel between the damping channel of Section 3 and the decision theory in Section 4 allows us to point out the role of discord in the noise resilience in the QI radar. This follows from the statement of the article [10]. Moreover, we used a link between both models that depends on the environment. This allows us to describe the environmental action on one hand and the signal-to-noise ratios depending on the thermal noise on another hand. In our toy model, we found that the discord is linked to the SNR calculated from the binary decision strategy. Consequently, it may be possible that the quantum discord is responsible of the quantum radar range from a quantum information point of view. Nevertheless, the presented toy model is unrealistic due to a lot of approximations, so we must discuss the limitations of such a model.

Firstly, we used a damping amplitude channel to model the propagation phase where we could use a master Equation [25]. However, the latter requires a lot of calculation based on an interaction Hamiltonian between the photon and the environment. This implies a good understanding of all interaction processes to define a suitable Hamiltonian for the entangled state. The damping amplitude channel is a sufficient tool of the quantum information theory to model the propagation phase because we consider the decoherence in an average process without detailing each phenomenon. The quantum channel approach gives an idea of the global evolution of the entangled state ${\hat{\rho }}_{\mathrm{AS},\mathrm{out}}$ in the atmosphere. A phenomenological approach would be better, but it is not in the scope of this paper.

Secondly, Lloyd’s binary decision strategy relies on approximations on thermal states in the small thermal noise $n_{b}$, so we are limited to the optical frequency regime for the QI radar. In addition, the link symbolized by $p=1-e^{-\gamma t}$ between the amplitude damping quantum channel and the decision strategy is very simple. Moreover, it would be difficult to set a realistic value to γ to improve the toy model. We do not have an exact correspondence between the separable product states obtained with the amplitude damping channel and with the binary decision strategy. However, the common thread between these models is that both states are product states over two maximally mixed qubit systems; hence, there are no quantum correlations remaining. The correspondence is not as good as expected because of the approximations on thermal states and because the quantum channel used is quite simple. It acts like a heat bath on the emitted qubit without introducing an average thermal noise while the thermal noise is introduced by Lloyd’s theory.

Thirdly, the model is realized for qubit states on the thermal energy levels, so for a photodetector that can see d=2 modes for an average thermal noise, $n_{b}\ll 1$. This allows us to link the model to the article [16], but this is an extremely limited case as the energy is very low due to the low number of thermal modes. Consequently, the model presented is not realistic, and in order to have a better approach, we should take a larger number of modes *d*. However, such a change also requires the modification of the amplitude damping channel used in Section 2.3. This is not as simple because the initial entangled state is more complex to define as for the quantum channel action on this state.

## 5. Conclusions

In this paper, we presented a toy model for a quantum illumination radar, focusing on the propagation channel that has a significant impact on radar detection. The QI radar uses pairs of entangled photons to perform a detection. We detailed the technical tools for a QI radar detection, and we introduced the quantum information theory to follow the evolution of quantum information. To highlight the environmental effect, we made a link between Lloyd’s binary decision theory and an amplitude damping channel modeling a propagation phase through a perturbative environment such as the atmosphere. Binary decision theory relies on the discrimination of quantum states corresponding to detection hypotheses $H_{0}$ and $H_{1}$. This decision theory has been adapted to the quantum channel used to perturb the quantum state of emitted photonic qubit S. The link between both models is performed using the damping parameter *p* acting on the thermal energy modes of the qubit propagating through the environment. The signal-to-noise ratios are calculated with the binary decision theory as functions of this damping probability *p*.

The calculations completed for the toy model suggest a link between the discord and sensitivity enhancement resilience in QI radar, as stated in [10]. In the amplitude damping channel based on the full thermalization of a photonic qubit, discord survived longer than entanglement. In the decision theory, the ${\mathrm{SNR}}_{+}$ vanishes only when p=1 corresponds to a vanishing discord. The disappearance of discord means the disappearance of quantum correlations inside the quantum system used for remote detection. The striking point in Lloyd’s theory was the sensitivity enhancement resilience despite entanglement lost. This resilience can be explained by discord because in both models, the limiting quantum state is a product state of the form ${\hat{I}}_{\mathrm{AS}}/d$, where *d* is the dimension of the quantum state. A product state is not entangled but, it could have quantum correlations in certain cases. Then, discord should partially explain the sensitivity resilience to decoherence produced by the propagation environment.

The toy model used is not a realistic model since it relies on many approximations, but it illustrates the role of quantum discord. In the quantum channel, we assumed the thermalization is fast and the only parameter is the damping probability *p*. In decision theory, we assumed a low thermal noise to make approximations for quantum states. Such an approximation on thermal noise is suitable for an optical frequency regime, but it does not work for a microwave frequency regime. In addition, we restricted ourselves to the limiting case of thermal qubits, which means we work with very low energy. To improve this model, we should work with photonic qudits, but that implies modifying the quantum channel used. Then, we have only considered single-photon emissions, which not very well adapted for radar detection in practice.

The results suggest further developing discord understanding for a QI radar. The current work takes part of a broader approach to the QI radar. It would be interesting to consider partially entangled states instead of fully entangled states. Moreover, a proper quantum radar would work with more modes and more photons to probe an object. To perform such a study, it becomes necessary to adopt the continuous variable formalism. Then, a quantum radar would work with microwave frequencies instead of optical frequencies, but it is theoretically and experimentally challenging because of the amount of noise in microwave frequencies. A possible way to overcome this difficulty is to increase the number of modes or the number of photons. At this stage of research, we have several ideas to explore.

This paper showed that the quantum discord explains the noise resilience of quantum radar, but we can think about a way to use it as for entanglement. The number of studies on quantum discord in quantum information process increased several years ago, as has the number of experiments on quantum illumination. We are at the very beginning of the development of quantum radar theory. This paper also provides way to spread the idea of the quantum radar potentiality among the radar community. However, please note that we cannot prove with certainty that the QI radar will surpass conventional radar. We can at least think of using both to improve radar detection in particular situations.

## Figures and Tables

**Figure 1 sensors-22-04821-f001:**
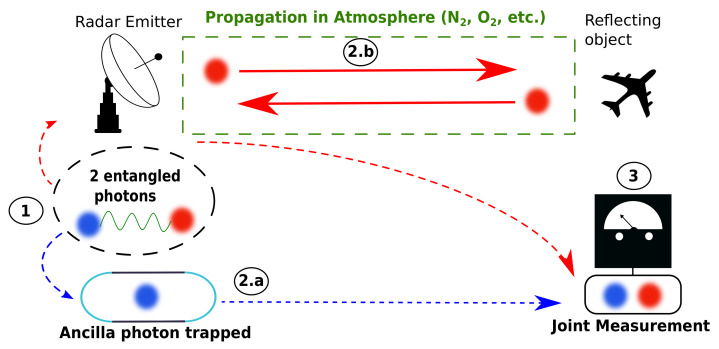
The quantum illumination radar process has three steps. In Step 1, the entangled pair of photons is created inside the radar. In Step 2, we perform a separation. In 2a, the blue photon is trapped inside the radar, while in step 2b, the red photon propagates through a medium before being reflected by the target. The red photon returns to the receiver. In step 3, the pair of photons is gathered, and a joint measurement of the quantum state blue–red is taken.

**Figure 2 sensors-22-04821-f002:**
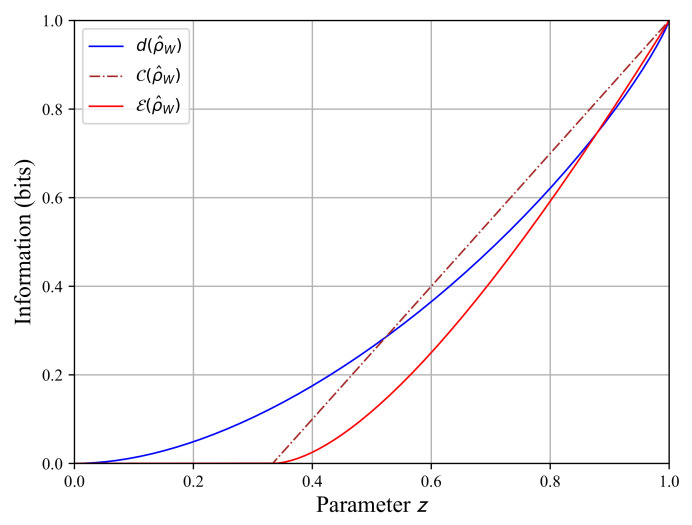
The entanglement rate and of the quantum discord as functions of the parameter z∈[0,1] for a Werner state ${\hat{\rho }}_{w}=z|\Psi^{-}\rangle {\langle \Psi^{-}|}_{\mathrm{AS}}+\left(1-z\right)/4{\hat{I}}_{\mathrm{AS}}$. The entanglement rate $E({\hat{\rho }}_{w})$ is in brown, and the concurrence $C({\hat{\rho }}_{w})$ is red. The quantum discord $d({\hat{\rho }}_{w})$ in blue represents the number of quantum correlations.

**Figure 3 sensors-22-04821-f003:**
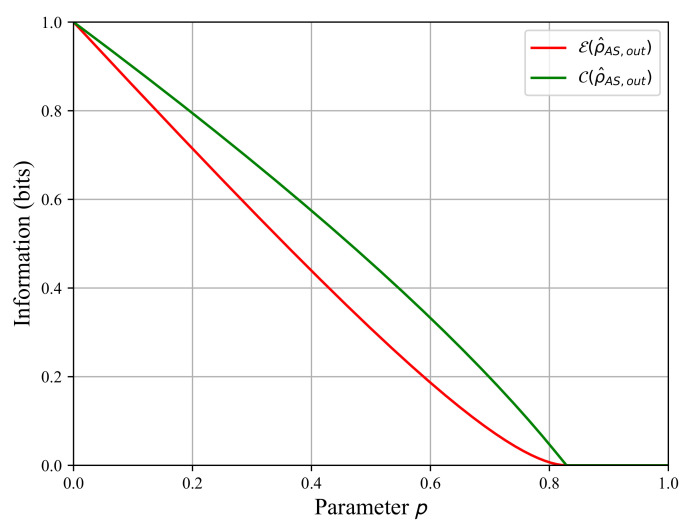
The concurrence of Wootters $C({\hat{\rho }}_{\mathrm{AS},\mathrm{out}})$ in green and the entanglement rate $E({\hat{\rho }}_{\mathrm{AS},\mathrm{out}})$ in red as functions of the parameter p∈[0,1] for ξ=1/2.

**Figure 4 sensors-22-04821-f004:**
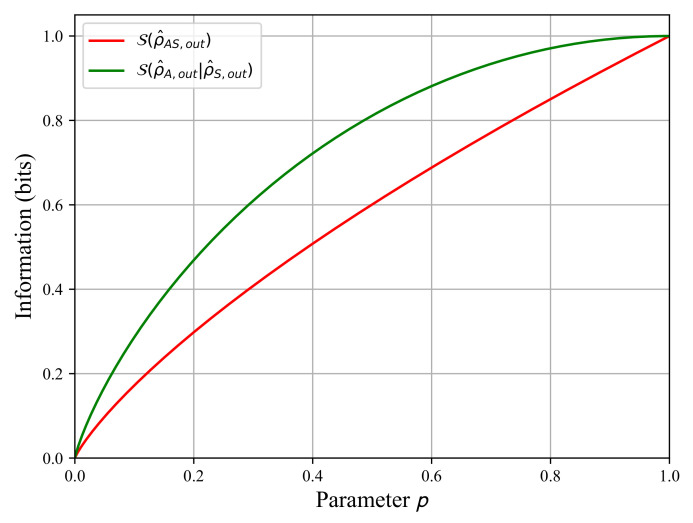
The Von Neumann entropy $S({\hat{\rho }}_{\mathrm{AS},}\;\mathrm{out})$ of the system AS in red and the conditional entropy $S{\left({\hat{\rho }}_{A,\mathrm{out}}|{\hat{\rho }}_{S,\mathrm{out}}\right)}_{\left\{{\hat{M}}_{S}^{\left(i\right)}\right\}}$ in green as functions of parameter p∈[0,1] for ξ=1/2.

**Figure 5 sensors-22-04821-f005:**
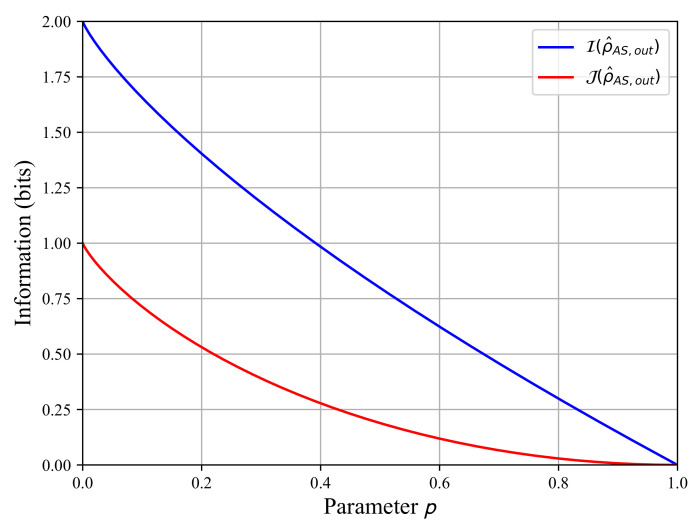
The quantum mutual information $I({\hat{\rho }}_{\mathrm{AS},\mathrm{out}})$ in blue and the classical information $J({\hat{\rho }}_{\mathrm{AS},\mathrm{out}})$ in red as functions of the parameter p∈[0,1] for ξ=1/2.

**Figure 6 sensors-22-04821-f006:**
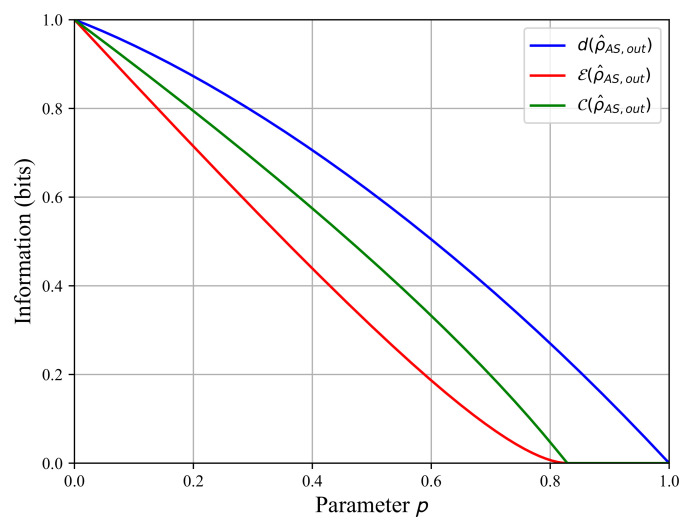
The entanglement rate $E({\hat{\rho }}_{\mathrm{AS},\mathrm{out}})$ (red), the concurrence of Wooters $C({\hat{\rho }}_{\mathrm{AS},\mathrm{out}})$ (green), and the quantum discord $d({\hat{\rho }}_{\mathrm{AS},\mathrm{out}})$ (blue) as functions of the parameter *p* for ξ=1/2.

**Figure 7 sensors-22-04821-f007:**
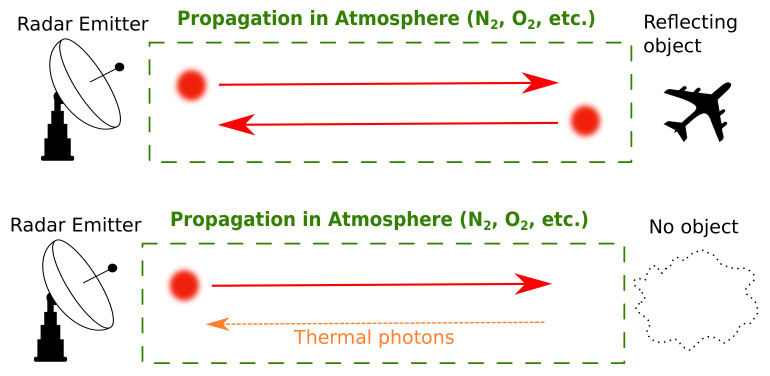
Detection strategies for the hypotheses $H_{0}$ and $H_{1}$ in Lloyd’s binary decision strategy for both single-photon and QI radars. When an object can be detected, we are in hypothesis $H_{1}$. In hypothesis $H_{0}$, only thermal photons can be detected, since no object is present.

**Figure 8 sensors-22-04821-f008:**
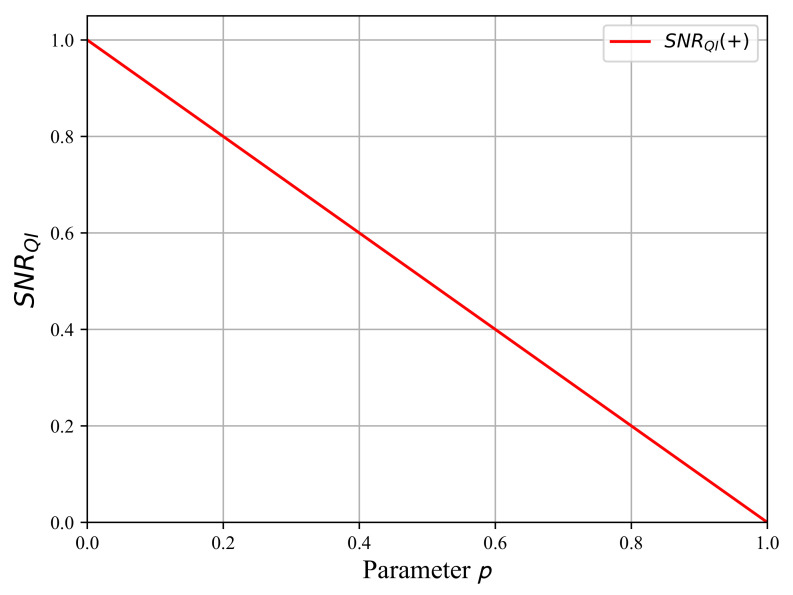
The ${\mathrm{SNR}}_{+}$ in the QI radar as a function of the parameter p∈[0,1] with $n_{b}$ calculated for $f=5\times 10^{14}$ Hz. The SNR has been normalized by its maximum because as the noise $n_{b}\ll 1$, the SNR is huge for small values of *p*.

**Table 1 sensors-22-04821-t001:** Probabilities of error for single-shot measurements for the single-photon radar and the quantum illumination radar.

Radar Scheme	Single-Photon		Entangled Photons	
**Outcomes**	**no = (−)**	**yes = (+)**	**no = (−)**	**yes = (+)**
$H_{0}$	$1-n_{b}$	$n_{b}$	$1-\frac{n_{b}}{2}$	$\frac{n_{b}}{2}$
$H_{1}$	$p(1-n_{b})$	$\left(1-p\right)+pn_{b}$	$p(1-\frac{n_{b}}{2})$	$\left(1-p\right)+p\frac{n_{b}}{2}$

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
