# Peer review of "Estimation of the Influence of a Noisy Environment on the Binary Decision Strategy in a Quantum Illumination Radar"

_sensors, 2022, doi:10.3390/s22134821_

Round 1

Reviewer 1 Report

The author claim that they presented a toy model for a quantum illumination radar focusing attention only on the propagation channel that has a significant impact on radar detection. QI radar uses pairs of entangled photons to perform a detection. To highlight the environment effect, they made a link between Lloyd’s binary decision theory and an amplitude damping channel modeling a propagation phase through a perturbative environment like the atmosphere. The calculations completed in the toy model suggest a link between the discord and sensitivity enhancement resilience in QI radar. Despite it is unrealistic, deeply understanding the hidden character of quantum illumination is still important. I recommend this manuscript to be published in this journal.

Reviewer 2 Report

The paper "Estimation of the Influence of a Noisy Environment on the Binary Decision Strategy in a Quantum Illumination Radar" presented a toy model to emphasize the great relations between discord and the sensitivity enhancement resilience for quantum illumination radar under the amplitude damping channel. However, looking at the manuscript, it's not clear offhand that it's "very important." As stated in the paper, Lloyd’s article had been proved that the sensitivity enhancement induced by the entanglement could be maintained even if the entanglement is lost during the propagation phase. Besides entanglement, discord which demonstrates the general quantum correlations beyond entanglement had been studied in many literatures. Discord is, of necessity, the origin of the advantages existing in quantum radar.

It is also hard to follow why the amplitude damping channel can be used to simulate the decoherence for qubit S evolved under the atmosphere, since it is never elaborated in detail. Moreover, in its current form, the presentation has so many problems. For example:

  1. The font should be unified. For example, the subscript of “t” in Eq. (1) is presented in italic. But it is different in main text.
  2. In Eq. (2), P_p is the probability of the photon not being absorbed and scattered by atmospheric molecules. However, there is also an assumption that photon is not absorbed or scattered by the propagation environment above Eq. (2). The assumption of simply reflected by the object in Step 2a combined with a perturbation due to the interaction with a distant object in Step 2b.
  3. “We will now…” above Eq. (3).
  4. Abbreviations are not explained. Such as “QI”, “EW”…
  5. “in view of to the decision theory” above section 2.
  6. For Eq. (15), the authors state that there is no object to reflect the photon in Hypothesis H0 and the object is present in Hypothesis H1, which is opposite to Fig. 7.

An important goal of manuscripts is to clarify ideas to the readers, and unfortunately this work is not successful on this. It seems to be poorly conceived, and the confusing explanations do not help the reader. This is quite regrettable and I feel sorry about this. I am sure that the researchers worked very hard on this, and they wish this to be published, but the final manuscript must clarify ideas, not confuse these.

I therefore cannot recommend the publication of this manuscript. 

Reviewer 3 Report

Borderieux et al. present a minimal model study on "Estimation of the Influence of a Noisy Environment on the Binary Decision Strategy in a Quantum Illumination Radar." Although it is a simple two photons entangled model, it shows an interesting correlation between the discord and sensitivity enhancement resilience in QI radar. I believe that the manuscript adds a new aspect of quantum illumination radar to the people in the quantum information science community and therefore is worthy of publication in the sensor MDPI journal. I recommend the manuscript for publication with minor grammatical checks and mathematical expressions in the final version.

Reviewer 4 Report

In this contribution, Borderieux and co-workers  have presented a simple, toy-model, for introducing an illumination radar, based on pairs of entangled photons to achieve the detection. In particular, the environment effect has been amply analyzed by introducing Lloyd's binary decision theory, suitably adapted to the quantum channel. Although the proposed toy model has interesting features, the authors also pointed out the many approximations that have been made. I found the paper well written, the theory sound, and for these reasons I recommend its publication in sensors. 

I have only a comment that I'd like the authors further develop. The strongest limitation of the proposed model is the fact that only single-photon emissions have been considered, which indeed makes the radar detection not possible in practice. I would like that the authors propose a possible improvement of their toy model, to take into account multiple photon emissions. 

Round 2

Reviewer 2 Report

The authors have improved their manuscript and clarified all my questions. I agree the publication of this manuscript in "Sensor".